# Enhanced Discrimination of Seismic Geological Channels Based on Multi-Trace Variational Mode Decomposition

**Jiaxuan Leng [1], Zhichao Yu [2,\*] and Chaodong Wu [1]**

[1] School of Earth and Space Sciences, Peking University, Beijing 100871, China; jxleng@pku.edu.cn (J.L.); cdwu@pku.edu.cn (C.W.)

[2] National Supercomputing Center in Shenzhen (Shenzhen Cloud Computing Center), Shenzhen 518005, China

\* Correspondence: zcyu.426@163.com

**Abstract:** The spectral decomposition is a valuable tool for improving the resolution of seismic interpretation, and thus can improve the accuracy of the subtle geo-features (thin and narrow channels, thin reservoirs, etc.). Variational mode decomposition (VMD) is an adaptive signal decomposition algorithm that non-recursively decomposes multicomponent signals into several band-limited intrinsic mode functions, which is competitive in enhancing time-frequency resolution. However, discontinuity is normally caused by the trace-by-trace process, making the 3D seismic interpretation difficult. To address this issue, we present a novel seismic geological channel detection method for 3D seismic dataset based on multi-trace variational mode decomposition (MTVMD). The proposed method decomposes the broadband seismic data into several intrinsic mode functions and then computes seismic attributes from each component for geological feature analysis. Further tested by field seismic case, the proposed method demonstrates strength in depicting the detailed edges and sedimentary signatures of paleochannels. Overall, the proposed method provides an alternative approach to identifying seismic channels, especially for the detailed portrayals of subtle geo-features in low-quality seismic data.

**Keywords:** seismic interpretation; spectral decomposition; seismic geological channel; variational mode decomposition; multi-variate variational mode decomposition

## 1. Introduction

Channel sand bodies are of great importance in many petroliferous reservoirs and are targets in oil and gas explorations [1]. The appropriate distribution of sand bodies, particularly sand thicknesses, aids in the investigation of sedimentary facies [2]. Therefore, the precise interpretation of seismic geological channels is highly demanding. Ascribing to the continuous channel migration and frequent vertical superposition of sand bodies, sedimentary patterns of the channels in seismic records are always complicated. Doubly inhibited by varied physical properties and oil-bearing properties on seismic data, traditional attribute-based analytical methods have difficulties in recognition, especially for identifying the precise edges and geological patterns of paleo-channels [3,4].

With spectral decomposition techniques, initial broadband seismic data can be decomposed into a sequence of frequency-decomposed seismic volumes of various frequencies. Separating the frequency bands of seismic data is theoretically valid for extracting the information on varied geological scales [5]. The spectral characteristics of relatively narrow band components are usually used to delineate the anomalous geologic features that would otherwise be hidden within the full-bandwidth seismic response; thus, appropriate bands would effectively increase resolution and reliability in seismic interpretation [6,7]. Accordingly, spectral decomposition-based methods were developed and widely applied in seismic data analysis, including sequence stratigraphy analysis [8], reservoir characterization [9], thin-layer thickness estimation [10], and hydrocarbon detection [11]. Conventional

spectral decomposition methods include short-time Fourier transform (STFT) [12], continuous wavelet transform (CWT) [13], and S transform (ST) [14]. The STFT, CWT, and ST are all bound by the Heisenberg uncertainty principle with a tradeoff between time and frequency localization [15]. In order to identify the intrinsic hidden properties contained in the nonlinear and non-stationary signals, the empirical mode decomposition (EMD) method is introduced by Huang et al. as a data-driven signal decomposition technique without using predefined wavelet libraries or base functions [16]. The EMD-based time-frequency analysis of seismic data shows the relative high resolutions in both the temporal and frequent realms, as it is effective on the instantaneous attributes of nonlinear and nonstationary signals. However, with a deficiency on the bandlimited assumption of the instinct mode function, the usage of the EMD method is limited. Further, the variational mode decomposition (VMD) method was developed, which can decompose the seismic signals into a sequence of quasi-orthogonal intrinsic mode functions, with the most compact bandwidth [17].

The VMD method was applied in seismic data analysis, including denoising [18,19], sequence stratigraphy analysis [7,20], seismic time-frequency analysis [21], and seismic facies analysis [22,23]. The VMD-based method, which calculates trace-by-trace, is unable to maintain the lateral continuity of 2D or 3D seismic data, causing difficulties in subtle geological feature discriminations [7], and consequently, including the seismic channels. Rehman and Aftab recently proposed the multivariate variational mode decomposition (MVMD) that is valid for multivariate or multichannel datasets [24]. In the MVMD method, the lateral continuity of the decomposed intrinsic mode functions (IMFs) is improved by applying lateral constraints, which are valid for extracting the spatial aspects of seismic reflection on different geological scales [25].

The tangled geo-features ascribed to the subtle discontinuities have brought difficulties in recognition by the original variational mode decomposition-based method. Taking into account the horizontal consistency of 3D seismic data, we propose a multi-trace variational mode decomposition-based seismic spectral decomposition method for geological feature discrimination in this study. First, we introduce the theory of variational mode decomposition and multivariate variational mode decomposition methods. Then, we illustrate the proposed MTVMD-based method for geological channel detection. Finally, we demonstrate the applicability of the proposed method to 3D field data.

## 2. Methods

In this section, we describe the variational mode decomposition method and multivariate variational mode decomposition method, then demonstrate the workflow of a multi-trace variational mode decomposition-based approach for 3D seismic data interpretation.

### 2.1. Variational Mode Decomposition

The variational mode decomposition method aims to decompose a real valued signal into a series of sub-signals with specific sparsity properties of its bandwidth in the spectral domain, while reproducing the input [17]. It is an adaptive and non-recursive signal decomposition approach in which each mode is assumed to be an amplitude modulation-frequency modulation (AM-FM) signal with a bandwidth and corresponding center frequency, i.e.:

$$u_k(t) = A_k(t)\cos(\phi_k(t)),\tag{1}$$

where, $A_k(t)$ and $\phi_k(t)$ are the instantaneous amplitude and phase of $u_k(t)$ respectively. The whole process of VMD can be considered as the construction and solution of a constrained variational problem, as described by Equation (2):

$$\min\left\{\sum_{k=1}^{K}\left\|\partial_t\left[\left(\delta(t)+\frac{i}{\pi t}\right)\times u_k(t)\right]e^{-i\omega_k t}\right\|_2^2\right\},$$
$$s.t.\quad\sum_{k=1}^{K}u_k(t)=x(t)\ ,\tag{2}$$

where $u_k$ is the $k$th decomposed mode, $\omega_k$ is the center frequency of the corresponding $k$th decomposed mode, $\delta(t)$ is a Dirichlet function, $x(t)$ is the input signal. The term $(\delta(t) + i/\pi t) \times u_k(t)$ is the Hilbert transform of $u_k$.

A quadratic penalty term $\alpha$ and Lagrangian multipliers $\lambda$ are adopted to render the variational problem unconstrained; the quadratic penalty term can guarantee better convergence properties, even in the presence of Gaussian noise, and the Lagrangian multiplier is used to enforce a strict constraint. The augmented Lagrangian is described as follows:

$$
\begin{aligned}
L(u_k, \omega_k, \lambda) = \alpha \sum_{k=1}^{K} \left\| \partial_t \left[ \left( \delta(t) + \frac{i}{\pi t} \right) \times u_k(t) \right] e^{-i\omega_k t} \right\|_2^2 \\
+ \left\| x(t) - \sum_{k=1}^{K} u_k(t) \right\|_2^2 + \left\langle \lambda, x(t) - \sum_{k=1}^{K} u_k(t) \right\rangle .
\end{aligned}
\tag{3}
$$

where $\alpha$ is a parameter balancing the variational energy term and the fidelity term. The alternating direction method of the multipliers (ADMM) can be employed to solve Equation (3). The modes and the center frequencies are updated iteratively by Equations (4) and (5), respectively:

$$
\hat{u}_k^{n+1} = \frac{\hat{x} - \sum_{j \neq k} \hat{u}_j + (\hat{\lambda}/2)}{1 + 2\alpha(\omega - \omega_k)^2}.
\tag{4}
$$

$$
\omega_k^{n+1} = \frac{\int_0^\infty \omega |\hat{u}_k(\omega)|^2 d\omega}{\int_0^\infty |\hat{u}_k(\omega)|^2 d\omega}.
\tag{5}
$$

where $(\hat{\cdot})$ indicates a Fourier-transformed signal. The Lagrangian multiplier will also be updated according to the following equation after each epoch:

$$
\lambda^{n+1} = \lambda^n + \tau \left( \hat{x} - \sum_k \hat{u}_k^{n+1} \right).
\tag{6}
$$

Finally, the above iteration terminates when the convergence condition is satisfied, namely:

$$
\sum_k \|\hat{u}_k^{n+1} - \hat{u}_k^n\|_2^2 / \|\hat{u}_k^n\|_2^2 < \varepsilon.
\tag{7}
$$

According to the preceding descriptions, there are four parameters that must be specified in advance: the mode number $K$, the balancing parameter $\alpha$, the constraint intensity $\tau$, and the tolerance of convergence criterion $\varepsilon$. The detailed complete algorithm of VMD can be found in [17]. It is important to note that VMD decomposes the input signal trace by trace, the lateral continuity is not considered.

### 2.2. Multivariate Variational Mode Decomposition

The goal of multivariate variational mode decomposition (MVMD) is to analyze a multivariate input signal containing $M$ number of data channels into a predefined number $K$ of multivariate components. As an extension of the VMD method, $K$ multivariate components that will fully construct the input signal $\mathbf{x}(t)$ are searched:

$$
\mathbf{x}(t) = \sum_{k=1}^{K} \mathbf{u}_k(t).
\tag{8}
$$

where the $k$th multivariate component. $\mathbf{u}_k(t) = [u_{k,1}(t), u_{k,2}(t), \ldots, u_{k,M}(t)]$ is a vector with $M$ components.

MVMD takes the multivariate input signal as a whole and tries to seek $K$ number of multivariate components from the input signal, with a minimum sum of band-

widths [24]. The optimization problem for MVMD can be naturally expanded from Equation (2) as follows:

$$\min_{u_k, \omega_k} \left\{ \sum_{k=1}^{K} \sum_{m=1}^{M} \left\| \partial_t \left[ \left( \delta(t) + \frac{i}{\pi t} \right) \times u_{k,m}(t) \right] e^{-i\omega_k t} \right\|_2^2 \right\},$$

$$s.t. \quad \sum_{k=1}^{K} u_{k,m}(t) = x_i(t), i = 1, 2, \ldots, M. \tag{9}$$

In Equation (9), the target function becomes the sum of the bandwidth of all of the modes over all of the channels, and the central frequencies $\omega_k$ in each mode are the same among different channels. The target function in Equation (8) can be reformulated as an augmented Lagrangian:

$$L(u_k, \omega_k, \lambda) = \alpha \sum_{k=1}^{K} \sum_{m=1}^{M} \left\| \partial_t \left[ \left( \delta(t) + \frac{i}{\pi t} \right) \times u_{k,m}(t) \right] e^{-i\omega_k t} \right\|_2^2$$

$$+ \sum_{m=1}^{M} \left\| x_m(t) - \sum_{k=1}^{K} u_{k,m}(t) \right\|_2^2 + \sum_{m=1}^{M} \left\langle \lambda_m(t), x_m(t) - \sum_{k=1}^{K} u_{k,m}(t) \right\rangle. \tag{10}$$

where the Lagrangian multiplier and the penalty weighting factor play the same role as those in VMD. The ADMM procedure is also used for the solution of Equation (10), as formulated in Equations (10)–(12), which is slightly different from that in VMD:

$$\hat{u}_{k,m}^{n+1} = \frac{\hat{x}_m - \sum_{j \neq k} \hat{u}_{j,m} + \left( \hat{\lambda}_{l,m}/2 \right)}{1 + 2\alpha(\omega - \omega_k)^2}, \tag{11}$$

$$\omega_k^{n+1} = \frac{\sum_{m=1}^{M} \int_0^{\infty} \omega |\hat{u}_{k,m}(\omega)|^2 d\omega}{\sum_{m=1}^{M} \int_0^{\infty} |\hat{u}_{k,m}(\omega)|^2 d\omega}, \tag{12}$$

$$\lambda^{n+1} = \lambda^n + \tau \left( x_m(\omega) - \sum_k u_{k,m}^{n+1} \right), m = 1 : M \tag{13}$$

Until convergence:

$$\sum_m \sum_k \left\| \hat{u}_{k,m}^{n+1} - \hat{u}_{k,m}^n \right\|_2^2 / \left\| \hat{u}_{k,m}^n \right\|_2^2 < \varepsilon. \tag{14}$$

The associated explanation can be located in the original work of Rehman and Aftab [24] and is not repeated here.

Figure 1 gives a schematic illustration of the algorithm of MVMD for a 2D seismic profile. In the algorithm, MVMD unifies the quantity of the decomposed component and their center frequency. It can not only increase the seismic signal similarity, but also the calculation efficiency by exchanging the time through memory space. The decomposition results reflect the seismic geological structures on various scales.

### 2.3. Multi-Trace VMD Based Seismic Channel Detection Method

Inspired by multi-variational mode decomposition [24] and quasi-bivariate variational mode decomposition [26] methods, we proposed a multi-trace variational mode decomposition-based (MTVMD) approach to deal with a 3D seismic dataset with only one primary dimension with discernible quasi-periodicity. However, instead of a simple concatenation of these intrinsic mode functions (IMFs) among the second non-decomposed dimension (inline and crossline direction are assumed here), the carrier frequency of IMFs with the same rank should be rebalanced among all of the traces. Such a consideration is based on the idea that there should be no big inter-trace discontinuity in each IMF; therefore, this step serves as an additional constraint to minimize the mode mixing problem.

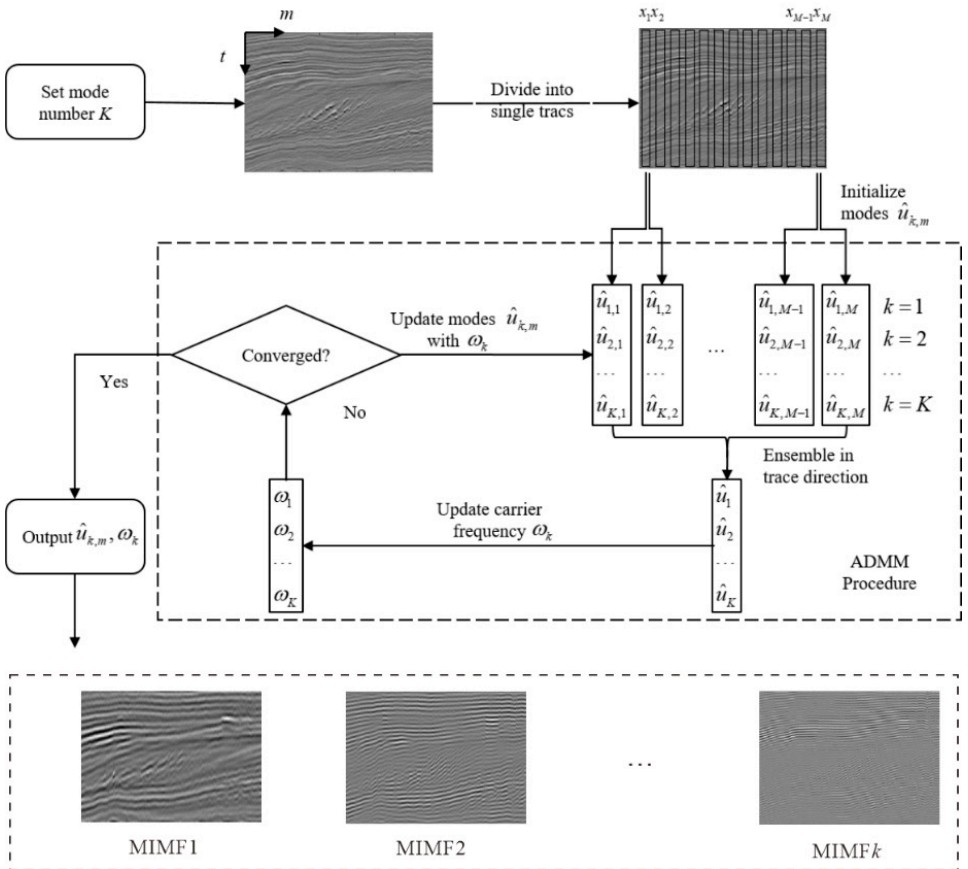

**Figure 1.** A flowchart of the MVMD algorithm for 2D seismic data.

The 3D seismic data $x_{l,m}(t)$ with $L$ number of inline traces and $M$ number of crossline traces is converted to be a quasi-2D seismic dataset after dimensionality reduction with the trace from both inline and crossline directions. We propose a novel seismic channel detection method, based on MTVMD in conjunction with seismic attributes. The detailed workflow (shown in Figure 2) can be summarized as follows.

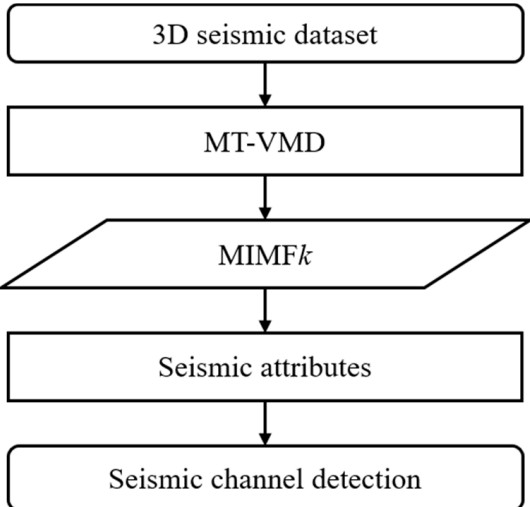

**Figure 2.** A flowchart of the proposed method for channel detection.

(1)  Decompose the 3D seismic dataset into a series of band-limited components using the MTVMD method;

(2)　The instantaneous frequency and amplitude of each component are utilized to obtain the seismic attributes, such as Root Mean Square amplitude, amplitude variance, and coherence;

(3)　The attributes are involved in the analysis of seismic geological feature interpretation, such as seismic channels.

## 3. Field Data Application

We evaluate the performance of the proposed method by a 3D field seismic data. The dataset here is derived from the Chepaizi uplift in the western Junggar Basin, China. The Chepaizi Uplift was formed during the deposition of the Neogene Shawan Formation as a large-scale terrestrial slope with fluvial-delta sedimentation in the western basin, and it currently offers a potential exploratory perspective [27,28]. These data contain the sedimentary transition from the braided channel to the curved channel and the near-end delta, with frequent vertical superposition of sand bodies and continuous channel migration in the Shawan Formation. Discriminating subtle geological features of sedimentary channels from seismic data is difficult, due to the complicated patterns of the fluvial-delta deposition. Figure 3 shows the 3D field dataset, and time slices and vertical profiles are extracted from the seismic volume. The exacted time slice at t = 1665 ms (Figure 4a), the vertical profiles at inline No. 98520 (Figure 4b), and crossline No. 11272 (Figure 4c) are used as cases. As seen in the vertical profiles, pinch out on both sides of the seismic reflections are characteristic of the fluvial seismic facies (noted by the black arrows), however, hardly seen features of an entire channel in the horizontal time slice.

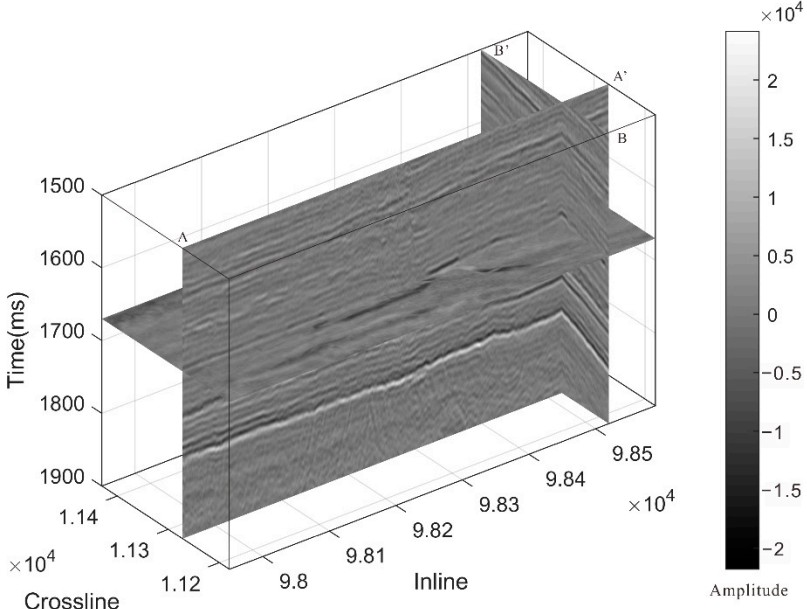

**Figure 3.** The 3D view of the field seismic dataset. AA' and BB' are through-well seismic profiles.

We can use the through-well seismic profiles to analyze more precisely the properties of the seismic geological channels. Figure 5 depicts the location of Well C1 on the southeastern channel of the selected real data. Through the lithological characteristics of the channel of Well C1, the grain size gradually becomes finer, the grains are coarser, and the lithology is gravel-bearing fine sandstone with a thickness of 2~9 m. The logging facies at the black ellipse dotted line is a zigzag box shape, which is in line with the logging facies type characteristics of the channel. On the well-passing seismic section, it can be seen that the seismic facies at the location of the black ellipse dotted line are continuous-moderately strong amplitude, which is consistent with the characteristics of the seismic facies type of the channel, and has a channel shape in the lateral direction.

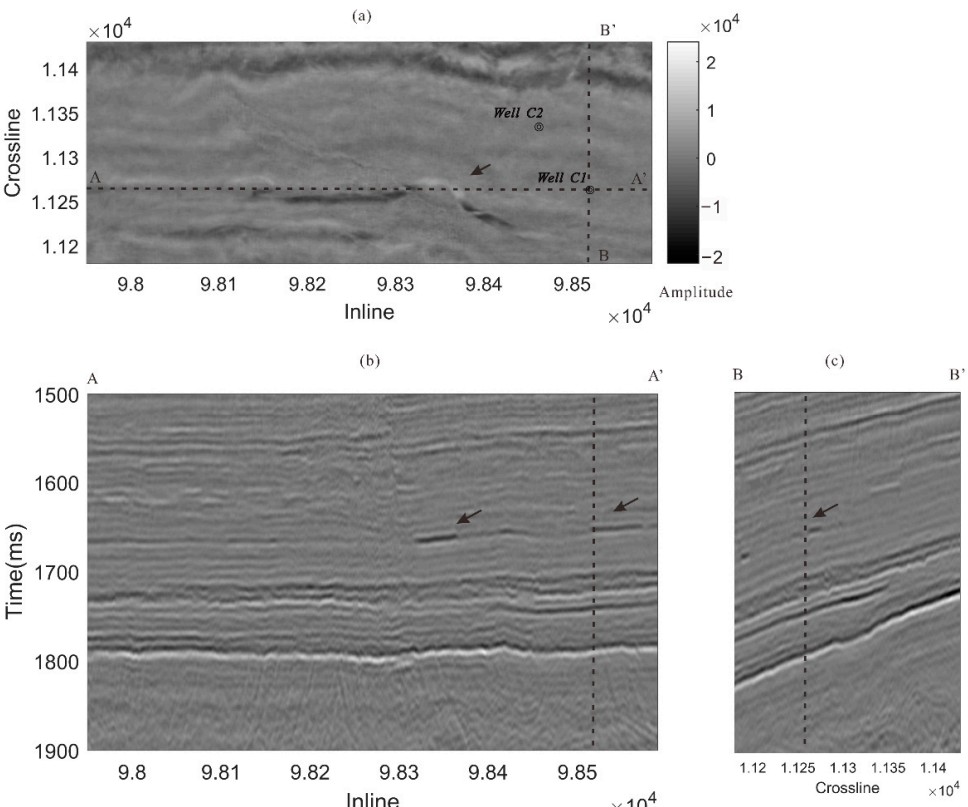

**Figure 4.** The location of wellbore and the through-well seismic profiles. (**a**) The time slice at t = 1650 ms; (**b**) the vertical seismic profile extracted at Crossline No. 98520; (**c**) the vertical seismic profile extracted at Crossline No. 11272. The dotted lines indicate the location of the through-well seismic profiles.

We use the full bandwidth dataset to calculate the Root Mean Square amplitude to analyze the seismic geological channels in this dataset, as shown in Figure 6. The figure shows that the extension direction and detailed description of the subtle channel facies are not clear. Apparently, spectral decomposition on the frequency bands is necessary in this case. We evaluate the performance of the proposed method by 3D field seismic data. To compare the performance on the resolution, we extracted band range frequency profiles and applied VMD- and MTVMD-based methods for comparison. Otherwise, RMS amplitude is else calculated. We demonstrate the superiority of MVMD over VMD using the through-well seismic profiles extracted from the seismic data. The input parameters $K$, $\alpha$, $\tau$ and $\varepsilon$ equal to 3, 2000, 0, and $10^{-4}$, respectively. Figures 7 and 8 depict the decomposed components by using the MTVMD and VMD methods on the vertical seismic profile AA' and BB', respectively. The center frequencies of the decomposed components are from low frequency to high frequency. In Figures 7d–f and 8d–f, we can observe that the lateral consistency of the 2D data cannot be guaranteed, because the processing is calculated trace by trace. Even a slight oscillation can completely alter the outcome of the breakdown, indicated by the black arrows. Instabilities such as these are a common drawback of the mode decomposition methods, which make it harder to control noise and cause structural abnormalities. Figures 7a–c and 8a–c show the multivariate intrinsic mode functions (MIMFs) from the MVMD method, geological structural changes are more consistent and reasonable, which improves seismic interpretation.

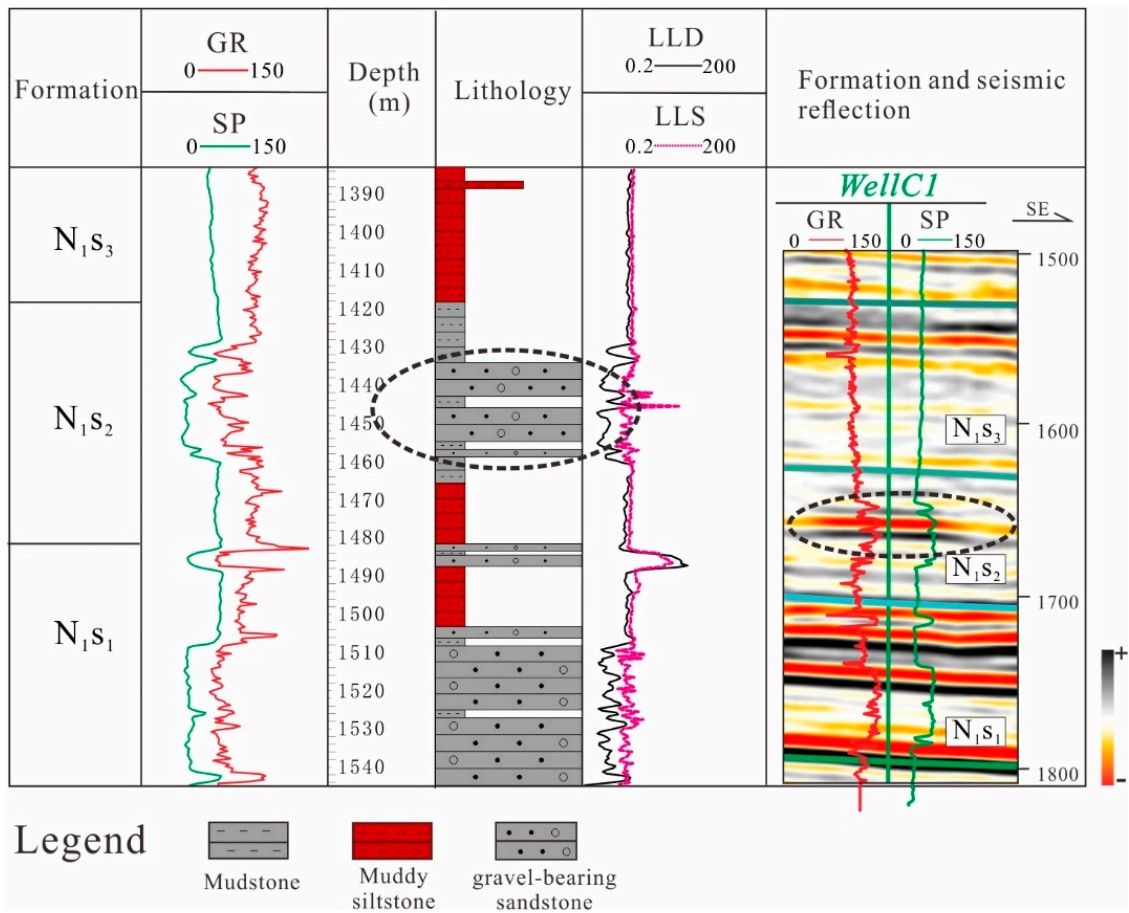

**Figure 5.** The well log recording of Well C1 in Figure 4. The dashed circle indicates the location of the seismic geological channel.

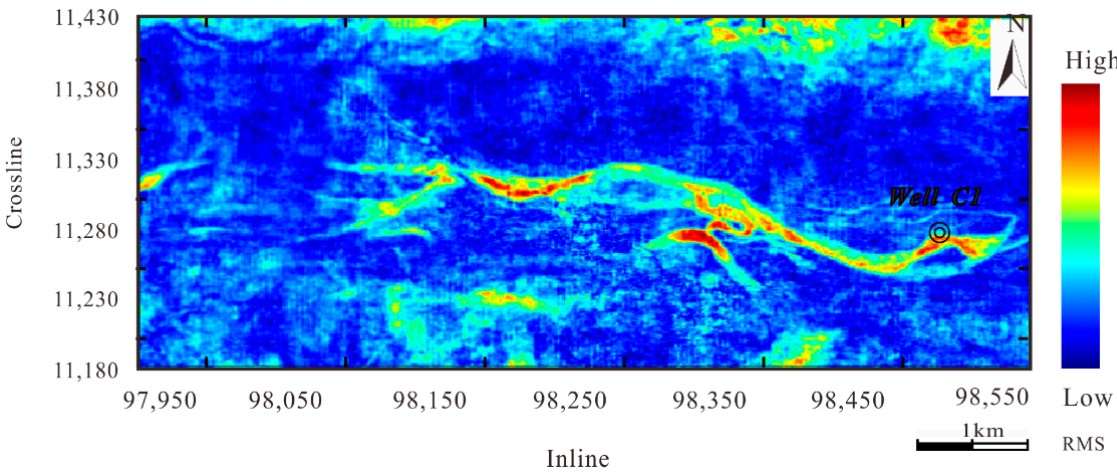

**Figure 6.** The RMS amplitude slices at t = 1650 ms calculated by full bandwidth seismic dataset.

For extracting the appropriate frequency bands, the dominant frequency of seismic data is considered, which is approximately 45 Hz on average with the effective bandwidth ranging from 10 to 81 Hz. Accordingly, three frequency bands of the RMS amplitude are chosen, including 10~33 Hz, 34~57 Hz, and 58~81 Hz, by the STFT method (Figure 9a–c). Consequently, three decomposed components are calculated by the VMD and MTVMD methods, as shown in Figures 10 and 11, respectively. Based on the VMD methods, channels of the fluvial-delta depositional system in the Shawan Formation are complete, verifying

that both the VMD and MTVMD methods are valid for enhancing the geological features' discrimination. However, details of the geological features, especially for the subtle seismic geological channel features, are dissimilar.

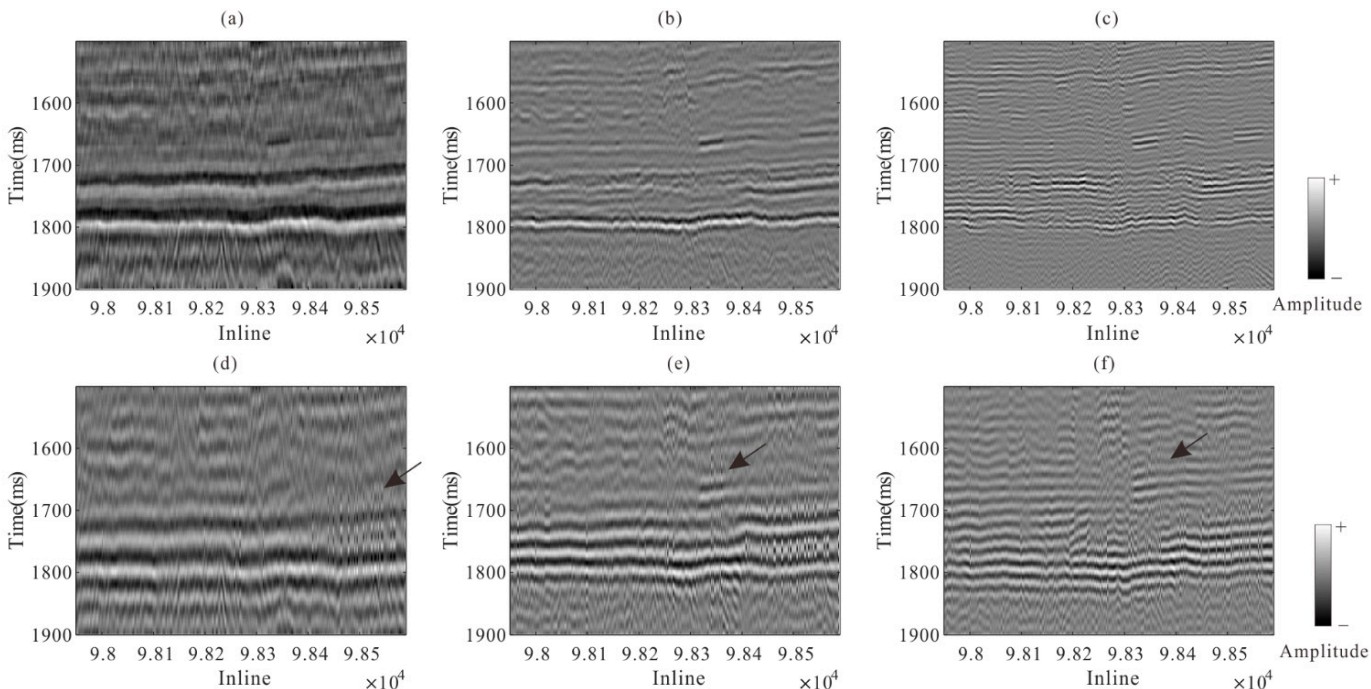

**Figure 7.** The decomposed components of the seismic vertical profile AA' by using the MTVMD and the VMD methods. (**a**–**c**) are the components from the MTVMD method; (**d**–**f**) are the components from the VMD method. Black arrows indicate the lateral inconsistency in seismic profile.

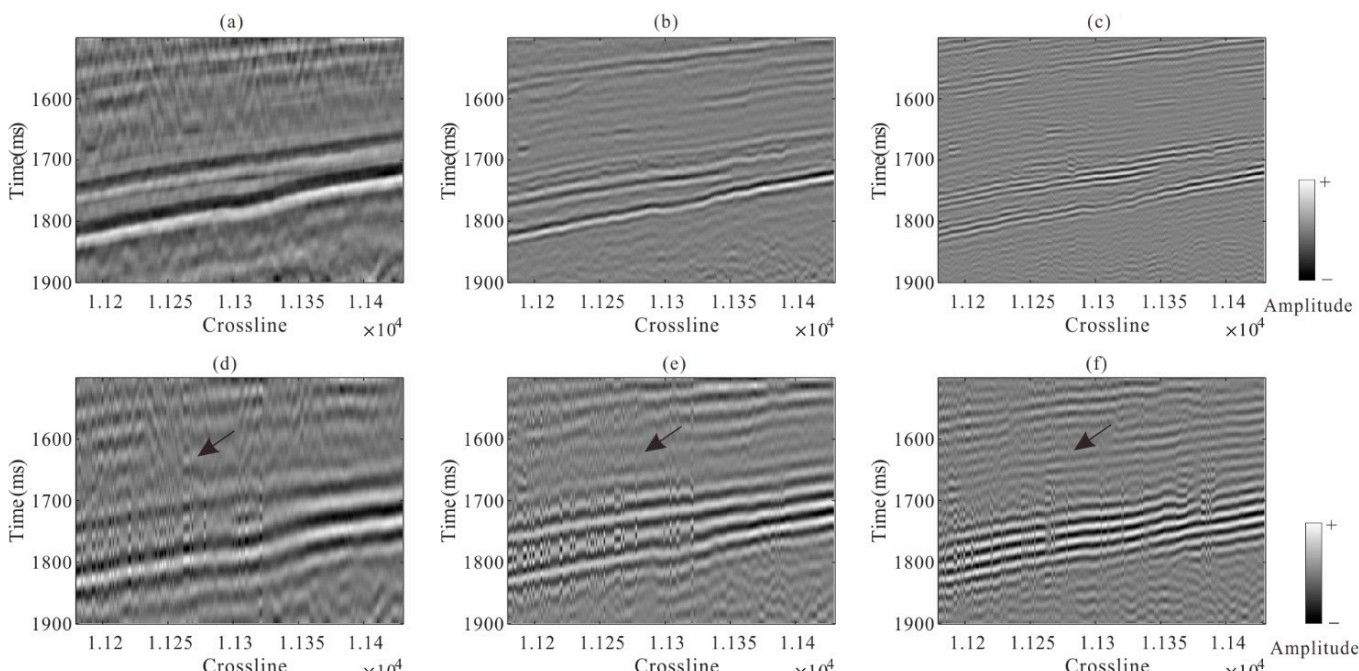

**Figure 8.** The decomposed components of the seismic vertical profile BB' by using the MTVMD and the VMD methods. (**a**–**c**) are the components from the MTVMD method; (**d**–**f**) are the components from the VMD method. Black arrows indicate the lateral inconsistency in seismic profile.

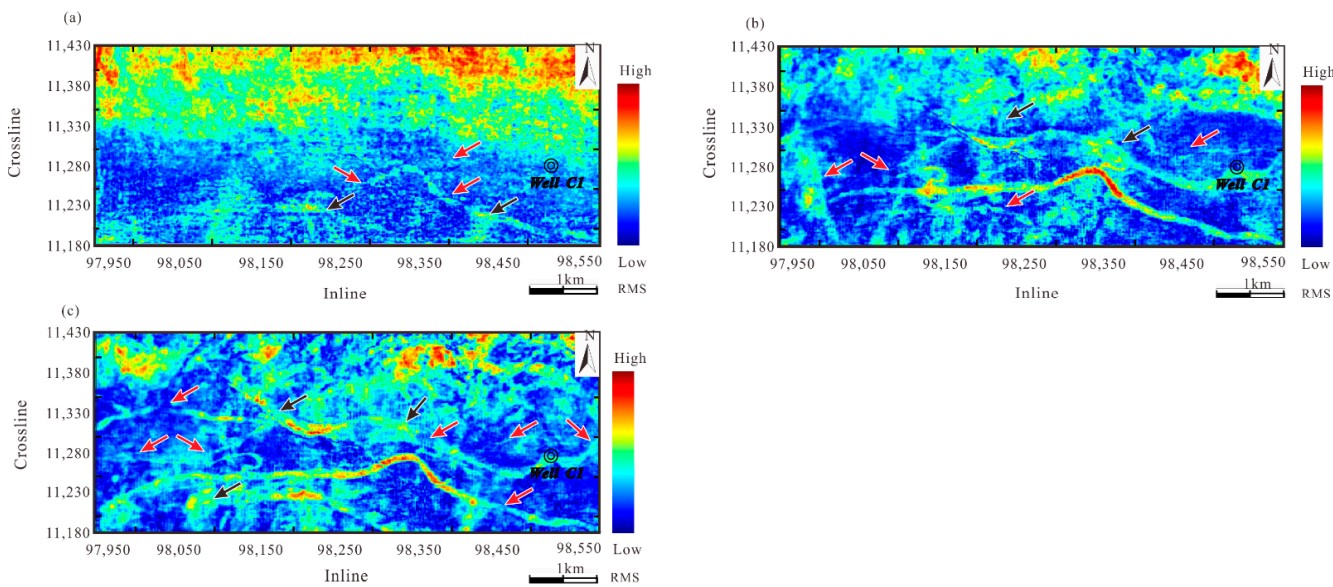

**Figure 9.** The RMS amplitude slices at t = 1650 ms. (**a**–**c**) show the RMS amplitude from the three frequency bands: (**a**) 10~33 Hz; (**b**) 34~57 Hz; (**c**) 58~81 Hz calculated by STFT method.

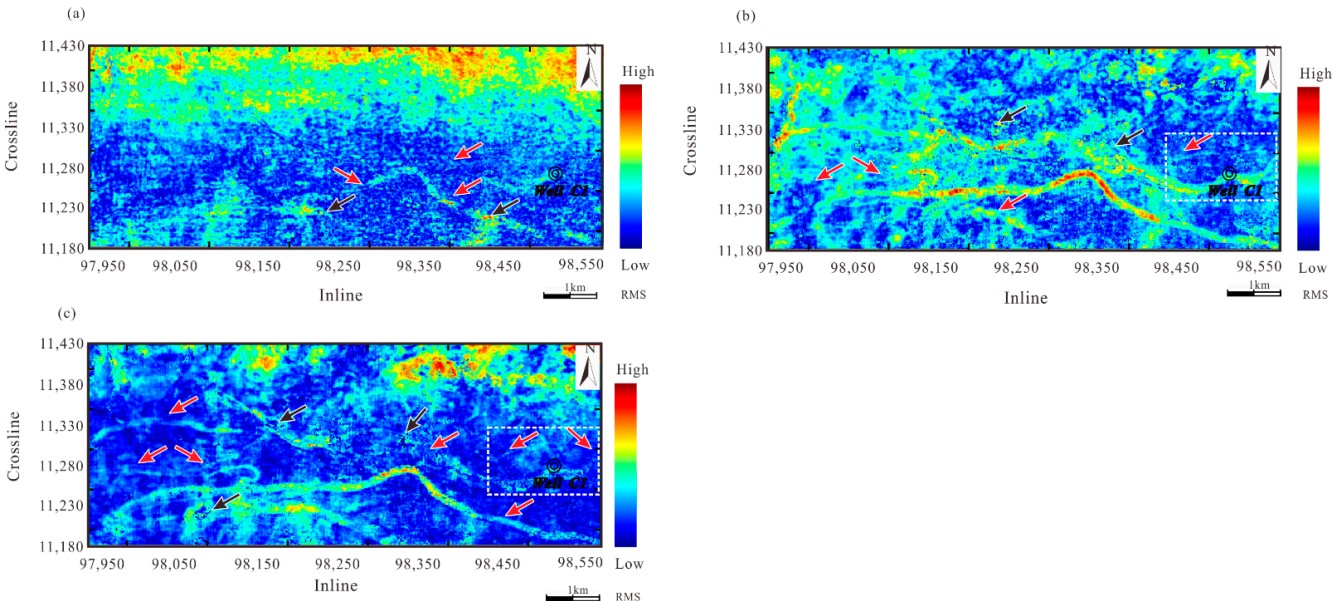

**Figure 10.** The RMS amplitude slices at t = 1650 ms. (**a**–**c**) show three decomposed components from the VMD method: (**a**) IMF1; (**b**) IMF2; (**c**) IMF3.

Recognized channels in fluvial-delta depositions are noted as red arrows in Figures 9–12. The presence of gaps and "jumps" in Figure 10a,c, marked as the black arrows, are evident due to the inconsistent center frequency. Overall, it is noticed that the high-frequency data (e.g., Figures 9c, 10c and 11c) are all presenting more sedimentary details compared to those of a low and medium frequency. However, it is always difficult to identify the spatial location, width, and flow direction of the narrow channels by using traditional time-frequency analysis methods. Although the high-frequency results are not as clear as the intermediate-frequency results in some locations, it is evident that the high-frequency spectrum is better described. For instance, some narrow channels (shown as the red arrow in Figure 11) in slices by the VMD method (Figures 10a and 11b) are expected but unrecognizable, however, these characteristics are significantly improved using the MTVMD based algorithm (Figure 11c,d).

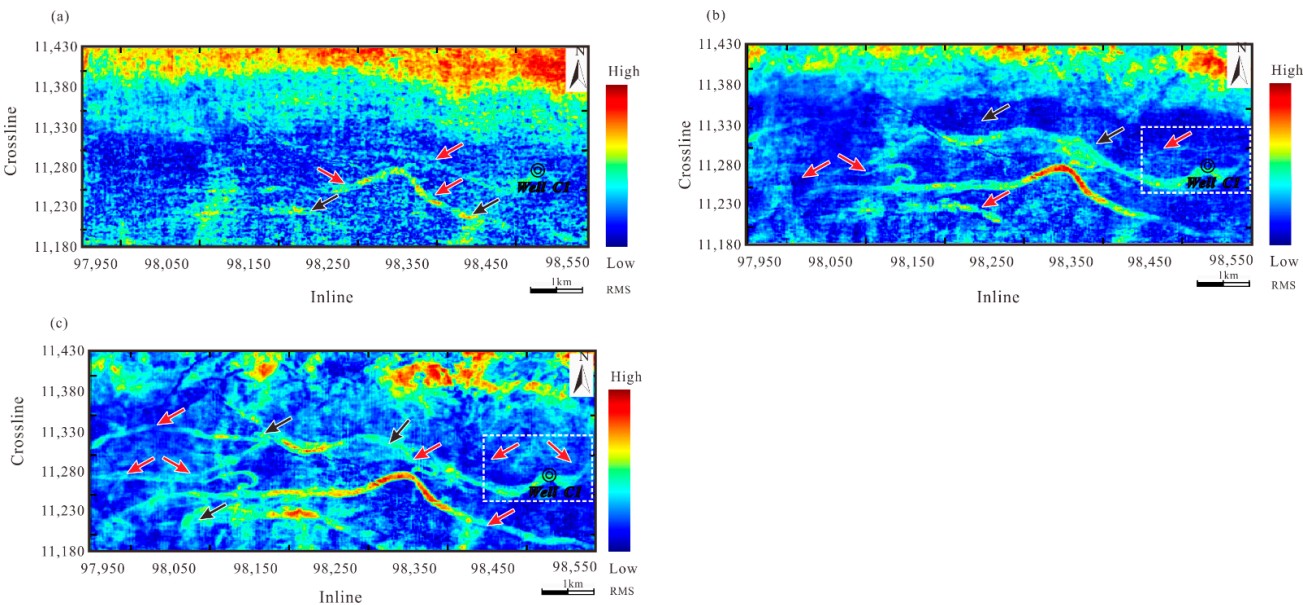

**Figure 11.** The RMS amplitude slices at t = 1650 ms. (**a–c**) shows three decomposed components from the MTVMD method: (**a**) IMF1; (**b**) IMF2; (**c**) IMF3.

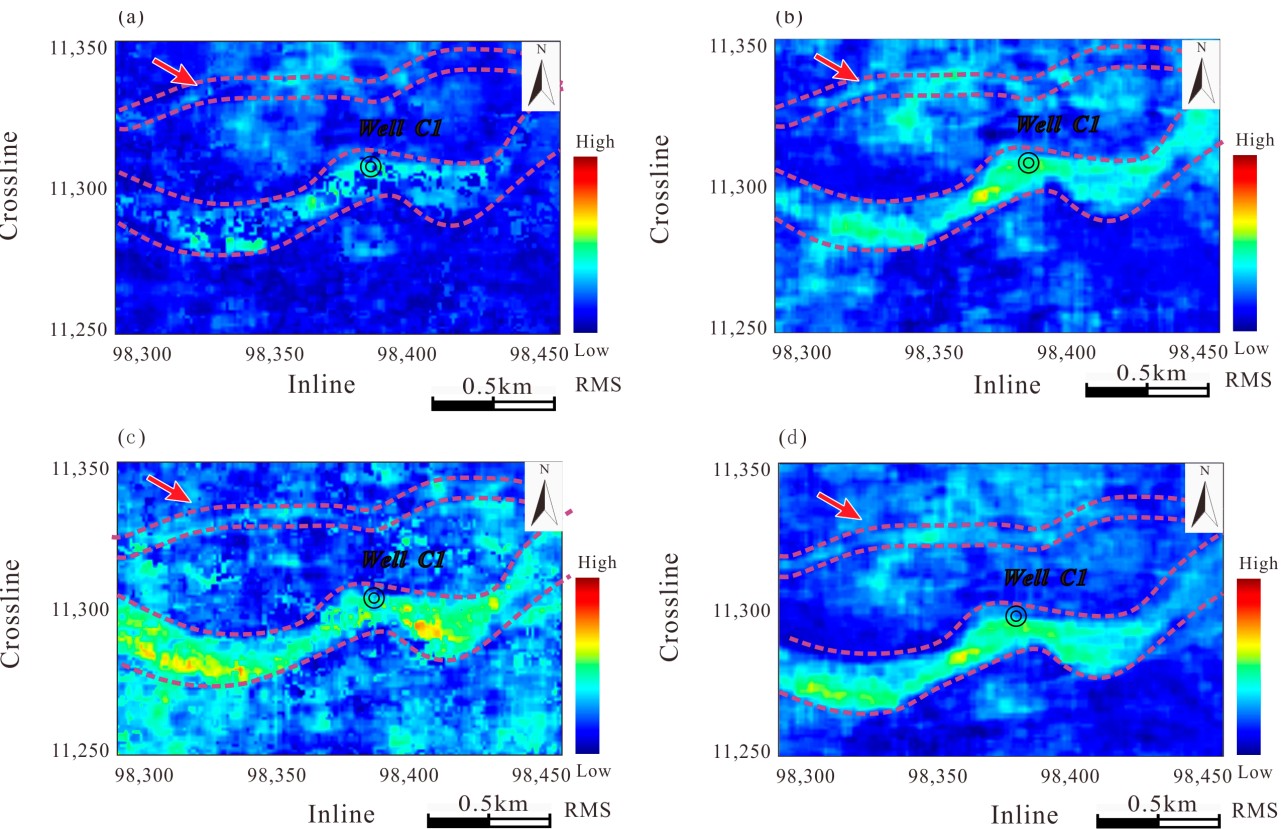

**Figure 12.** The detailed comparison in IMF2 and IMF3 components from the VMD and MT-VMD methods. Here, (**a,c**) are from Figure 10b,c, respectively; (**b,d**) are from Figure 11b,c, respectively. The original location of (**a–d**) are marked as white boxes in Figures 10 and 11.

Specifically, spatial spreading of the narrower branched channels is differentiated (Figure 10c, marked with a red arrow), except for the main stream of wide channels (Figure 9c). Subtle channels in lateral distributions are more distinct with the MTVMD method (Figure 11c, red dash lines). As estimated by scales, the resolution of channels by

the MTVMD method can reach around 16 m (e.g., 16 m thickness and 20 m width, as shown in Figure 11). In addition, by the MTVMD method, overall clear depositional patterns and lateral consistency are evident (Figure 11), with the comparison of the results decomposed by STFT and VMD (Figures 9 and 10), which is expected to result in more concentrated energy and reduced shadow effect from the dip of seismic data by the MTVMD method.

## 4. Conclusions

A novel multi-trace variational mode decomposition-based seismic spectral decomposition method is presented for the purpose of better discrimination of the subtle seismic geological channels. The conclusions from this study are that:

(1)　The variational mode decomposition method has the advantages of high resolution, complete decomposition, and avoidance of the mode stacking effect, which is suitable for enhancing the extraction of the target signal in seismic interpretation;

(2)　Multi-trace variational mode decomposition provides an improved lateral consistency and resolution for recognizing the subtle geological features of channels, which enhances the quality and reliability of seismic interpretation;

(3)　Tested by a 3D seismic dataset, the results have proved that, on data-driven decomposition approaches, a higher spectral-spatial resolution and clearer stratigraphy and lithology boundaries were obtained, compared to the traditional short-time Fourier transform method.

Although the computational efficiency was improved, a multi-core parallel computing strategy should be considered when processing 3D datasets by using the MTVMD method. With more big data input, the alternating direction method of multipliers for MVMD in the frequency domain may have high computation complexity. Data segmentation is also an effective strategy that divides the input data into a series of smaller ones, which can be considered for processing. In addition, subsequent research should also concentrate on the optimal selection of decomposition parameters.

**Author Contributions:** J.L. wrote the paper and performed the experiments; Z.Y. provided the original idea and wrote the program code; C.W. contributed to the conceptualization of the paper. All authors have read and agreed to the published version of the manuscript.

**Funding:** This research was financially supported by the National Science Foundation of China under Grant 42072125 and the National Science and Technology Major Project of China under Grant 2017ZX05008-001.

**Institutional Review Board Statement:** Not applicable.

**Informed Consent Statement:** Not applicable.

**Data Availability Statement:** Not applicable.

**Acknowledgments:** The calculations are supported by the High-performance Computing Platform of Peking University.

**Conflicts of Interest:** The authors declare no conflict of interest.

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
