# Peer review of "Enhanced Discrimination of Seismic Geological Channels Based on Multi-Trace Variational Mode Decomposition"

_applsci, doi:10.3390/app12115416_

Round 1

Reviewer 1 Report

A brief summary

The authors utilized 3D seismic reflection data from offshore Netherlands and China. This manuscript provided an alternative method on seismic interpretation area where the authors proposed Multi-Trace Variational Mode Decomposition (MT-VMD) method (in addition to conventional seismic attribute method) for sedimentary channels detection from low-quality 3D seismic reflection data.

General concept comments

  • Too many abbreviations on the text, so I need to scroll-up to its first-mention and it is time consuming to go back and forth.
  • Data and methods chapter is confusing, the authors use F3 seismic data on the method chapter and use other seismic data on the field data application chapter.
  • Need consistency through-out the manuscript, e.g., channels or river?
  • Channel’s interpretation on this manuscript is debatable since there is no wellbore data used in this study.
  • Notation on the Figures 5-9 is small with unsolid black colour, I need to zoom-in up to 400% on the PDF file to see the image clearly.

Specific comments

Line 2: I suggest to edit the terminology of “Seismic Channels”, since this might be confusing e.g., for non-applied geophysics. How about “sedimentary channels featured from seismic data”? (Actually, you’ve mentioned this on lines 21-22). Please edit this terminology through-out the manuscript.

Lines 2-3: As a seismic interpreter, this title is fine for me, but please keep in mind that MDPI Applied Sciences journal has very broad disciplines, so e.g., “Seismic Channels” might be confusing for seismologist. I suggest the authors to edit the title for accommodating very broad disciplines of this journal.

Line 12: Please edit the “geo-features” terminology on this line and through-out the manuscript, since this could be interpreted as “geo-logy”, “geo-physics”, geo-graphy”, etc. How about “geological featured”

Line 20: Is it possible to merge this sentence “Last, the attributes are involved in the analysis of geo-features” with the next or previous sentences, since this sentence is very short in a line.

Lines 21-22: “….in depicting the detailed edges and sedimentary signatures of paleochannels”. This is nice.

Line 31: Please add more geological terms on “stacking” and “migrations” terminologies, since these might be confusing for geophysicists.

 Line 72: I suggest to have “Data” chapters (Merge with chapter 2. Methods) on this manuscript, where the authors could explain what kind of data they used, vertical and horizontal resolution of the data, brief acquisition and processing workflows, phase and polarity of seismic data, data quality, seismic artefacts, etc. I’ve checked that the authors use F3 seismic data on this “method” chapter (See Figures 1-3), which is confusing to utilized data in method chapter.

Lines 122-126: Please edit these sentences to be more figure mentioning, not figure describing. Please apply through-out the manuscript.

In my opinion, describing figure should be on the figure caption. 

Lines 134-136: Figure 1, what kind of “slice” ?

Lines 137-138: Figure 2, What is the unit of the colour bars? The same question for other similar figure on the manuscript.

Lines 137-138: Figure 2, what is the horizontal distance among the seismic trace? In my opinion, the horizontal scale is needed. The same question for other similar figure on the manuscript.

Line 144: What kind of slices? Time slice? Horizon slice? Stratal slice?

Line 145: What do you mean by “inter-slice”,

Lines 158-159: In my opinion, definition of Root Mean Square amplitude, amplitude variance, and coherence is needed to describe on this manuscript. This is important since the authors still utilizing these seismic attributes to delineate the channels from 3D seismic data.

Line 164: Why the authors use F3 seismic reflection data on the method chapter and use different seismic data on the field data application chapter?

Line 164: Since there is no wellbore data on this study, how do you know that features marked by black and red arrows in Figure 5-9 are channels? I suggest to make one chapter about comparing your result with other well-known study to support your channel’s interpretation.

Lines 239-248: This looks like a conclusion, could be merge with conclusion chapter.

Line 249: Is it possible to have bullet points conclusion? It creates conclusion easier to read.

Author Response

We greatly appreciate all your good suggestions for the manuscript. Our manuscript has been revised carefully according to the comments and suggestions. Below are our responses to your comments.

Reviewer 2 Report

REVIEW

on article

Enhanced Discrimination of Seismic Channels Based on Multitrace Variational Mode Decomposition

Jiaxuan leng, Zhichao Yu, Chaodong Wu

SUMMARY

The article is devoted to solving an actual scientific and practical problem. The problem of complicated geostructures attributed to subtle heterogeneities and leading to difficulties in recognition by traditional analytical methods with low frequency, narrow bandwidth and low vertical and lateral resolution of seismic data is still relevant. The essence of the work was to create a new MT-VMD method with the aim of better recognition of subtle seismic geostructures.

The authors propose a new seismic channel detection method based on multi-trace variational mode decomposition (MT-VMD) for a 3D seismic dataset. Seismic field trials have been conducted, showing that the MT-VMD method demonstrates the ability to display paleochannel margins and sedimentary signatures in detail, as well as improve computational efficiency. The authors summarized that the proposed method provides an alternative approach to identifying seismic channels, especially for detailing subtle geostructures in low quality seismic data.

The results of the authors are original, practically significant and significant from the point of view of the scientific component.

In general, the article makes a positive impression, but needs significant improvement. The article can be published, but only after correcting all the comments below:

COMMENTS

  1. The author must redo the Abstract and bring it in compliance with the requirements of the Applied Sciences journal. The scientific problem is not described (Background). The scientific novelty is not indicated. The Abstract contains a rather long introductory part, the purpose of the study is also not formulated. Editors strongly encourage authors to use the following style of structured abstracts, but without headings: (1) Background: Place the question addressed in a broad context and highlight the purpose of the study; (2) Methods: Describe briefly the main methods or treatments applied; (3) Results: Summarize the article's main findings; and (4) Conclusions: Indicate the main conclusions or interpretations. The Abstract should be an objective representation of the article.
  2. The end of the Introduction section is logically to complete the formulated scientific novelty, as well as the purpose and objectives of the study.
  3. In addition, the work plan should have been presented in more detail, for example, in the form of a flowchart for better clarity and visualization of the author's idea.
  4. Why in Equation 1 do you consider the expansion of the function u(t) only in terms of even functions cos(t), while in Equation 2 there is a complex term containing odd functions as well?
  5. Line 105 vector not “vetor”.
  6. Check all figures for Applied Science journal requirements, 1000 pixels on the smaller side and 300 DPI resolution.
  7. The first mention of Figure 3 is located in subsection 2.2, and the figure itself is in subsection 2.3. It is necessary to move the figure closer to the first mention of it in subsection 2.2 or move the mention of figure 3 to subsection 2.3.
  8. At the end of subsection 2.2, after Figures 1 and 2, their interpretation should be added.
  9. It would be methodologically correct to finish Section 2 with a small analysis of the proposed method, and not with Figure 4. Considering comments 3-5, it is necessary to improve the structure of Section 2.
  10. Lines 233-235 should be more logically moved to the "Methods" section.
  11. It is strongly recommended to add some discussion to this study, that is, to compare the solutions proposed by the authors with those already known. It is also necessary to compare the obtained results with the results of studies presented in the "References" section. This would emphasize the novelty of this work and strengthen the manuscript.
  12. Smoother transitions should be added between sections to improve the structure of the article and its methodological component.
  13. The section "Conclusions" should be completed with information on the practical significance and possible directions for continuing this study.
  14. The section "References" should be increased by 5-10 sources of literature, mainly published in the last 5 years (2017-2022), due to a more detailed literature review and the addition of a section or subsection "Discussion". This would greatly strengthen the study in terms of its relevance and novelty.
  15. In the "References" section, hyperlinks to literature sources in the form of https://doi.org/ or doi numbers should be added for quick search and identification of works cited by the authors.
  16. There are some stylistic and speech inaccuracies in the work. It is recommended to check the text again and correct the shortcomings.
  17. English requires some adjustments. Needs to be checked.

Author Response

We appreciate all of your helpful suggestions for the manuscript. Our manuscript has been carefully revised according to the comments and suggestions. Below are our responses to your comments.

Round 2

Reviewer 2 Report

All my comments were considered and appropriate corrections were made in the article's text. The article looks much better. I recommend the article for publication.